# InSight: Self-Guided Skill Acquisition via Steerable VLAs

Maggie Wang[1], Lars Osterberg[1], Stephen Tian[1], Ola Shorinwa[2], Jiajun Wu[1], Mac Schwager[1]

[1]Stanford University    [2]Princeton University

https://insight-vla.github.io/

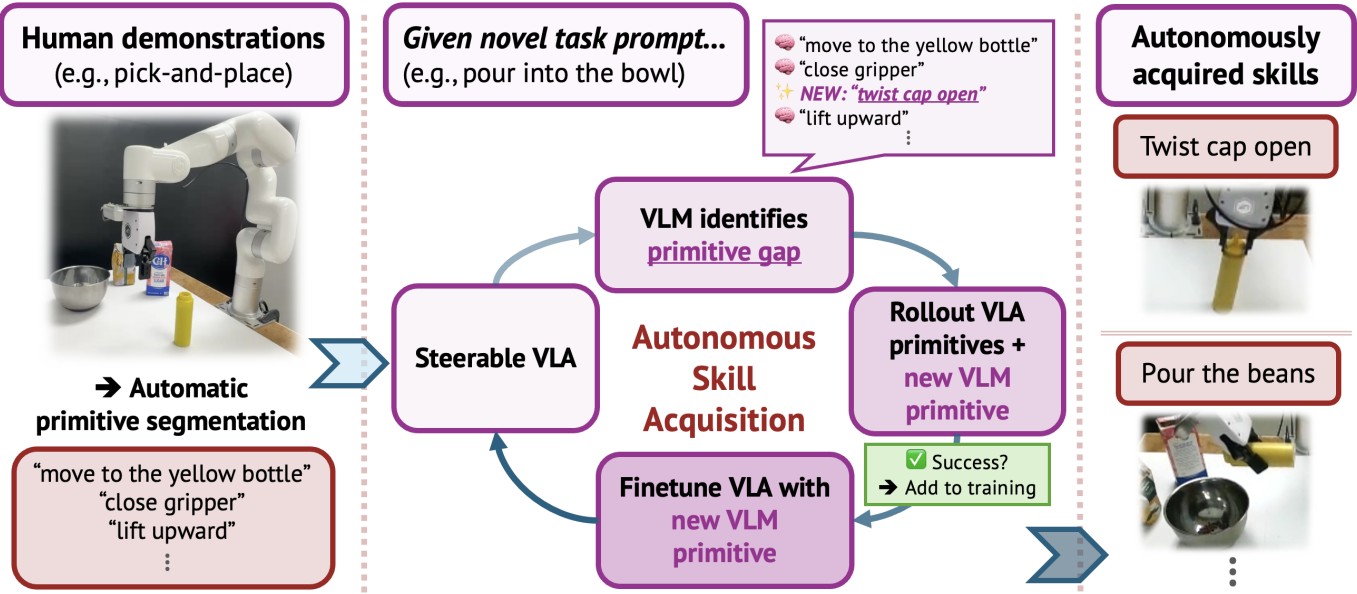

Fig. 1: **Overview of INSIGHT.** (1) Human demonstrations are automatically segmented into primitive-labeled trajectories to fine-tune a VLA to be steerable via these primitive labels. (2) Given a novel task, a VLM identifies missing primitives, autonomously collects successful rollouts, and retrains the VLA with the new primitives. (3) The newly acquired primitives (e.g., twisting and pouring) can be composed to learn new skills without additional human demonstrations.

*Abstract*—Vision-language-action (VLA) models can learn manipulation skills from demonstrations, but their capabilities are bounded by the skills in the training data. We present INSIGHT, a framework that unlocks autonomous skill acquisition by rendering VLAs steerable at the primitive-action level (e.g., "move gripper to the bowl", "lift upward", "pour the bottle"). INSIGHT consists of two primary stages: (1) an automated segmentation pipeline that partitions demonstrations into labeled primitives via VLM plan decomposition and end-effector poses to enable VLA primitive steerability, and (2) a VLM-guided data flywheel that identifies missing primitives required to accomplish a novel task, autonomously attempts demonstrations of the missing primitives with VLM-proposed low-level control, and automatically labels, stores, and integrates successful demonstrations into the VLA training set. We evaluate INSIGHT across simulation and real-world manipulation tasks, including block flipping, drawer closing, sweeping, twisting, and pouring, without any human demonstrations of these target skills. Once learned, these primitives can be composed to execute novel, long-horizon tasks without additional human demonstrations. Our findings demonstrate that primitive steerability provides a practical foundation for continual skill acquisition in VLA policies.

*Index Terms*—VLAs, VLMs, Skill Learning, Steerable Policies

## I. INTRODUCTION

Teaching robots new manipulation skills is expensive. Collecting human demonstrations and fine-tuning a policy requires substantial human effort for every new task. Vision-language-action (VLA) models have made progress toward general-purpose manipulation, but their capabilities remain bounded by the skills present in their training data [1, 2, 3].

Consider a robot operating on the surface of Mars that has been trained to scoop rocks for sample collection. If a dust storm deposits debris onto its solar panels, the robot may need to clear the panels [4], yet a VLA trained only to scoop may fail to execute the required sweeping behavior because no demonstrations of sweeping were provided. Acquiring new skills through interaction is also costly: simulation-based reinforcement learning (RL) often requires thousands of trials, while real-world RL largely remains impractical due to sample complexity and safety constraints [5, 6].

Our key insight is that manipulation skills are inherently compositional: new manipulation skills are rarely fully novel, but instead reuse previously seen primitives in new combi-

nations [7, 8]. For instance, sweeping and scooping share approach and lowering primitives, but differ in the lateral pushing primitive. Similarly, flipping a block reuses the same grasp-and-lift sequence from pick-and-place but adds a rotation primitive not present in those demonstrations. This structure suggests that existing VLAs may already encode reusable primitives, but these primitives are not easily *steerable* because they are entangled within the full task instruction [9, 10].

Efficiently acquiring a new skill requires not only executing primitives, which a VLA can be made to do, but also recognizing what primitives are missing to achieve a task, which a vision-language model (VLM) can provide. While recent work uses VLMs and LLMs as planners, coding agents, or trajectory-generation modules that compose pre-trained or pre-defined primitives at test time [11, 12, 13, 14], these methods extend the robot's behavior through test-time reasoning without updating the learned policy. We propose a different role for the VLM: not only as a test-time planner over existing skills, but as an *active agent* for identifying *missing* primitives, generating successful robot rollouts, and adding those rollouts back to the VLA by retraining to extend its skill capabilities.

This process is analogous to how humans encounter a novel scenario: we understand what skills we can already perform, and thus recognize when current skills are insufficient. We then reason about what new capability would bridge the gap and learn using targeted practice. The acquired skill can then be stored as a reusable capability for future tasks, thus enabling continual, lifelong learning.

We propose INSIGHT, a framework for **open-world skill acquisition via steerable VLAs**. Figure 1 summarizes the overall INSIGHT pipeline, from primitive segmentation of human demonstrations to VLM-guided acquisition and modular composition of new primitives. We show how a VLA can be made steerable at the level of composable manipulation primitives and then autonomously extended when a novel task requires a missing primitive. Our contributions are as follows:

- An **automatic primitive segmentation** pipeline that decomposes teleoperated demonstrations into labeled primitives without manual annotation, enabling primitive-level VLA steerability.
- A **VLM-guided primitive acquisition** loop that identifies missing primitives for novel tasks, executes them with VLM-derived parameters, and retrains the VLA on autonomously generated demonstrations to accomplish new skills.

We validate INSIGHT across five tasks in simulation and on hardware, including block flipping, drawer closing, sweeping, twisting, and pouring. We demonstrate that our framework enables autonomous skill acquisition with zero target-skill human demonstrations, achieving up to 96% success on tasks such as pouring, and 80% success on a complex 14-primitive long-horizon task while retaining full performance on original base skills.

## II. RELATED WORK

**Vision-Language-Action Models and Steerable Policies.** VLAs map visual observations and language instructions to robot actions, with policies such as OpenVLA [1] and $\pi_{0.5}$ [2] learning end-to-end control from language-labeled robot data. Recent work explores finer-grained language conditioning for more steerable control interfaces [15, 9]. STEER [10] shows that dense language relabeling can expose an expressive low-level control interface, while Steerable Policies [9] similarly uses language-conditioned primitives to guide behavior at test time. VLS [16] is a training-free steering framework that uses a VLM to synthesize reward functions that guide a pretrained diffusion policy toward out-of-distribution spatial and task configurations without retraining. These methods establish steerability as a useful control interface, but treat the resulting primitive set as fixed. In contrast, INSIGHT uses steerability as the foundation for skill acquisition. Given a steerable VLA with a set of known primitives, INSIGHT identifies missing primitives for a new task, generates successful rollouts for those primitives, and adds them back into the VLA's training data. This expands steerability from a test-time control interface into a mechanism for persistently expanding the policy's reusable skill set.

**Skill Decomposition and Composition.** Hierarchical robot learning methods decompose manipulation into reusable skills that can be sequenced for long-horizon tasks [17, 18, 19, 7]. Bottom-up skill discovery methods extract reusable primitives from unsegmented demonstrations or reward-free interaction using clustering, representation learning, hierarchical imitation learning, or factorized skill spaces [20, 21, 22, 23]. Unlike methods that assume a fixed set of primitives and skills, INSIGHT couples skill decomposition with autonomous discovery to extract primitives from demonstrations and acquire new primitives with VLM guidance.

**LLM and VLM-Guided Robotics.** A growing body of work uses foundation models to provide high-level semantic guidance for robot execution. SayCan [12] grounds LLM action proposals with value functions over pretrained skills, but it does not learn missing low-level skills. VoxPoser [13] uses LLMs and VLMs to build 3D value maps, but also does not expand the learned primitive library. Code-as-Policies [11] uses LLMs to compose program-like skills at test time, while Hi Robot [24] uses hierarchical VLAs to interpret complex instructions and incorporate feedback. While these methods plan over existing primitives, they operate at test time: the robot may perform a new task through reasoning or composition, but the underlying learned policy is not expanded. INSIGHT instead uses the VLM as part of a data acquisition loop that identifies and acquires missing primitives to accomplish novel skills.

**Autonomous Skill Acquisition.** Reinforcement learning (RL) has studied how agents can acquire skills through interaction, but in real-world manipulation, RL typically demands dense rewards, large amounts of real-world interaction data, and a narrow sim-to-real gap [25], which are generally challenging to achieve. Recent work reduces human supervision

through foundation model-generated rewards [26, 27, 28], automated demonstration data [29, 30], or language-labeled trajectories [31] for robot policy learning. While these methods also focus on reducing human supervision for skill acquisition, they typically operate at the level of rewards, demonstrations, or full-task trajectories. In contrast, INSIGHT focuses on a smaller unit of acquisition: we acquire missing primitives that can be reused compositionally across various tasks.

**Continual, Lifelong Skill Learning.** A related line of work studies how robots can continually acquire new skills while retaining prior skills [32]. Recent methods address this through structured knowledge reuse: Stellar VLA [33] uses a continually evolving knowledge space with expert routing, while SkillsCrafter [34] separates skills into distinct semantic skill subspaces. INSIGHT takes a data-centric approach by retraining a single VLA jointly on the original and newly acquired primitives, so existing primitives remain supervised as the vocabulary grows.

## III. SKILL ACQUISITION VIA STEERABLE PRIMITIVES

INSIGHT operates in two stages: (1) training a steerable VLA on automatically segmented primitives, and (2) autonomous skill acquisition, where a VLM orchestrates the full loop of primitive gap identification, data generation, success evaluation, and retraining. Figure 2 provides an overview: Figure 2a shows the automatic primitive segmentation stage, and Figure 2b shows the VLM-guided skill acquisition loop.

### A. Stage 1: VLA with Steerable Primitives

#### 1) Primitive and Skill Definition

In this paper, we define a **skill** as a target capability described by a language instruction (e.g., *unscrew the bottle cap and pour the contents into the bowl*). A **plan** is the sequence of primitives that the VLM planner generates to complete a skill.

A **primitive** is a reusable action segment that the VLA produces when conditioned on its language label. Following the precondition formalism of task and motion planning (TAMP) [8], each primitive is characterized by a *precondition* on the world state where it is invoked and an *effect* on the resulting state. In our setting, primitives are designed to have a single dominant motion mode (e.g., translation or rotation along an axis or a gripper transition). Each primitive is associated with a natural-language label that describes its semantic effect (*move gripper to the yellow bottle*, *lift upward*, *twist*, *pour*). The VLA executes each primitive end-to-end until a termination signal fires (e.g., a learned progress detector crosses a completion threshold).

Let $\mathcal{V}$ be the policy's primitive vocabulary, or the set of primitive labels for which the VLA has been trained. Given a plan $\mathcal{P} = (p_1, \ldots, p_n)$ generated by the VLM planner for a skill $s$, a *primitive gap* is any $p_i \in \mathcal{P}$ such that $p_i \notin \mathcal{V}$. INSIGHT autonomously acquires successful rollouts for each primitive gap. After these rollouts are added to the training set and the VLA is retrained, $\mathcal{V}$ expands and future plans can invoke the acquired primitives as known capabilities rather

## Stage 1 (of 2): Automatic Primitive Segmentation for VLA Steerability

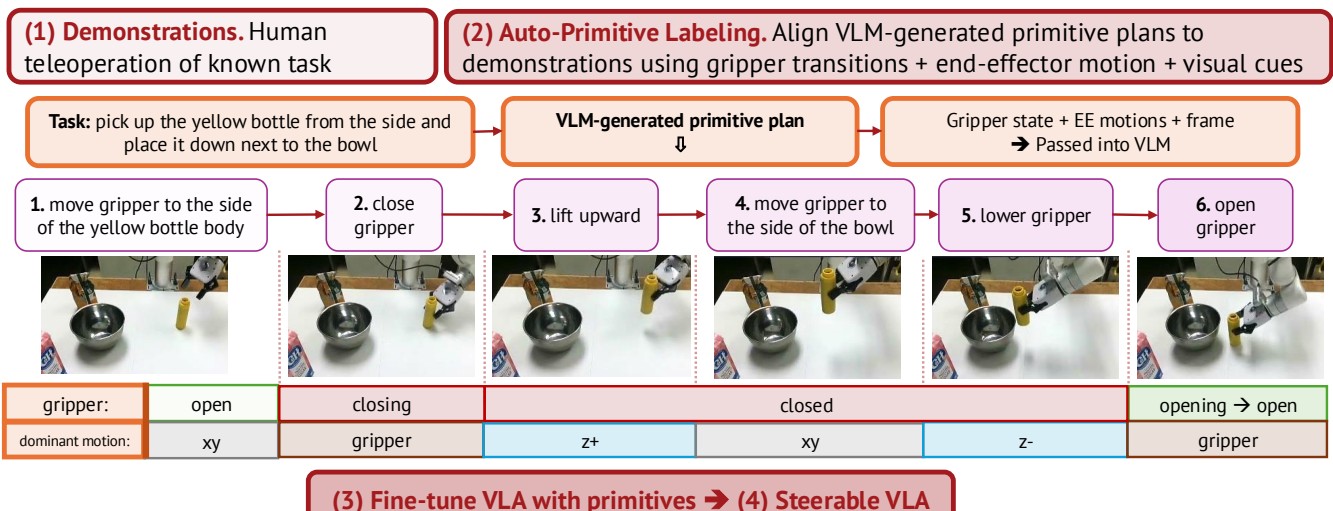

(a) **Stage 1: automatic primitive segmentation.** (1) Human collects teleoperation data of a known task. (2) These demonstrations are decomposed into labeled primitives by aligning a VLM-generated primitive plan with gripper-state transitions and dominant end-effector motion. (3) The primitive-labeled trajectories are used to fine-tune a VLA. (4) Individual primitives can be composed for language-conditioned steerability in new tasks. The example shows a side pick-and-place demonstration segmented into six primitives, with the gripper-state and dominant-motion signals used for alignment shown below the frames.

Fig. 2: **INSIGHT overview.** (a) Stage 1 builds a steerable VLA from primitive-segmented demonstrations. (b) Stage 2 (next page) uses a VLM to identify and acquire missing primitives for novel tasks, adding successful rollouts back into the VLA.

## Stage 2 (of 2): VLM-Guided Skill Acquisition

**(1) Novel task.** Pour the yellow bottle into the bowl

**(2) Identify primitive gap.** Given known primitives, create plan

**(3) Execute rollout.**

**Known: 1.** move to bottle, **2.** close gripper, **3.** lift up, **4.** move to bowl, **7.** lower, **8.** open gripper

**VLM proposes gap parameters:**
- **axis** {dx, dy, dz} (translational) or {drx, dry, drz} (rotational)
- **magnitude** = θ or cm

**1.** tilt bottle forward  **(+dry)**     **2.** tilt bottle back upright  **(-dry)**

**Primitive gaps in plan:**
**5.** tilt bottle forward
**6.** tilt bottle back upright

VLM oracle success check ➔ Success? ➔ Add to dataset

**(4) Retrain with new primitive**

(b) **Stage 2: VLM-guided skill acquisition.** (1) Given a novel task, (2) the VLM builds a plan and flags any primitive missing from the VLA's current vocabulary as a *primitive gap*. (3) During a rollout, known primitives are executed by the steerable VLA, while each primitive gap is executed by a low-level controller parameterized by a VLM-proposed motion axis and signed magnitude. (4) A VLM oracle verifies task success, and the successful rollouts of each new primitive are added to the training set to fine-tune the VLA with the acquired primitive. In this example, INSIGHT acquires the *tilt bottle forward to pour* and *tilt bottle back upright* primitives needed to pour the bottle into the bowl.

Fig. 2: **INSIGHT overview** (continued). Stage 2 of the INSIGHT pipeline.

than primitive gaps. This framework enables new skills to be realized without additional human demonstrations.

*2) Automatic Primitive Segmentation*

As shown in Figure 2a, we automatically segment teleoperated demonstrations into labeled primitives without manual annotation.

Given a task description, a VLM produces an ordered sequence of expected primitives. We give the VLM a set of example primitives (e.g., "close gripper", "lift upward") to provide examples of the granularity of primitives. The primitive set is extended whenever the VLM proposes a new primitive needed to complete the task.

For tasks involving a gripper, we segment boundaries at gripper open and close transitions from the gripper command velocity. The end-effector pose, derived motion magnitudes (in $xy$ and $z$), and a dominant-axis tag (one of $\{xy, z, rxy, rz\}$) are passed to the VLM to match each frame with its associated primitive name.

The segmented primitives are used to fine-tune a pretrained VLA using LoRA [35] (additional details in Appendix A). Each primitive segment becomes a separate training episode, conditioned on its primitive label as the language prompt. To provide a primitive-level termination signal, we add a learned progress channel to the action space, with progress labels in $[0, 1)$ within each primitive.

*B. Stage 2: VLM-Guided Skill Acquisition*

Given a steerable VLA trained on a base set of primitives, INSIGHT autonomously expands the skill set when presented with a novel task that requires missing primitives. A flowchart of this system is shown in Figure 2b.

First, the VLM decomposes the task into a primitive sequence and compares against the known primitive vocabulary.

Primitives not in the vocabulary are flagged as **primitive gaps**. The planner is constrained to return one single-axis motion per primitive gap. Therefore, tasks requiring multiple distinct motions (e.g., tilt forward and then tilt back) produce multiple primitive gaps rather than a single composite primitive.

Next, each primitive gap is parameterized by an axis (translation $\{dx, dy, dz\}$ or rotation $\{drx, dry, drz\}$) and a signed magnitude. A pre-execution VLM call analyzes the current scene and proposes the parameters. For chained gaps within a single rollout, the prior gap's parameters are passed to the VLM to ensure a consistent axis and magnitude for paired motions (e.g., pour-forward then tilt-back-upright).

To acquire a new primitive, each plan execution produces a sequence of (known and/or new) primitives, which is rolled out by the robot. A post-plan VLM oracle compares the scene images before and after skill $s$ is run to judge full task success. Successful new primitives within $s$ are added to the VLA training dataset. After $n$ rollouts of the skill, we combine the existing primitive vocabulary $\mathcal{V}$ with the new primitive(s) $p$ to retrain the VLA. The retrained model can execute the full composed task autonomously, with the VLM acting only as a high-level planner to sequence the primitives.

## IV. EXPERIMENTS AND ANALYSES

We evaluate INSIGHT across simulation and real-world manipulation tasks designed to test whether primitive-level steerability can support efficient skill acquisition, out-of-distribution (OOD) execution, and compositional reuse.

In simulation, we use a 7DoF Franka Panda in the LIBERO [36] environment to study block flipping from pick-and-place demonstrations to measure how performance improves as autonomously acquired rotate-block primitives are added, and drawer closing from drawer opening demonstra-

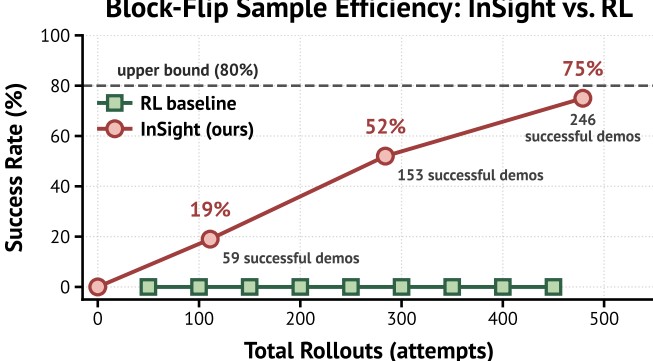

Fig. 3: **Block flip sample efficiency: INSIGHT vs. RL.** Full flip success rate as a function of total environment rollouts (task attempts), with the number of rotate-block primitives in grey. The RL SAC [37] baseline (given the same rollout budget) does not complete a flip (0%), although it learns to reach the block (in 23% of episodes) and grasp it (in 10% of episodes), but never lifts and rotates it to completion. The upper bound success is shown by the dotted line at 80%.

tions to test whether INSIGHT can acquire a missing primitive from an OOD initial state.

On hardware, we use a 6DoF UFactory xArm to evaluate bottle twisting and pouring to compare against a Code-as-Policies-style zero-shot baseline [14], and then compose the separately acquired twist and pour primitives along with the base pick-and-place skills into a long-horizon twist-then-pour task. We measure whether the unified policy retains its original pick-and-place skills after new primitives are added. Finally, we evaluate whether INSIGHT extends to contact-rich, non-prehensile motions by acquiring a sweeping primitive from scooping demonstrations.

### A. Simulation: Block Flipping from Only Pick-and-Place Human Demos

For this task, the robot is asked to flip a Lego block such that the peg is facing right side up, given only human demonstrations of block pick-and-place. The desired new skill picks up a block tilted on its side, flips it right-side-up, and places it down on the table.

We collect 150 human teleoperated pick-and-place demonstrations, where the block is on its side. We automatically segment these demos into over 700 primitive episodes across seven primitive types. The block-flip task requires a *rotate-block* primitive that is not present in pick-and-place demonstrations, and the VLM identifies it as a primitive gap.

**In this setting, bootstrapping skill acquisition with primitives is more sample-efficient than our RL baseline, suggesting that INSIGHT can be a practical method for real-world skill acquisition.** Figure 3 shows that block-flip success improves as rotate-block primitive rollouts are added, reaching 75% success after 246 acquired primitive rollouts, collected over 479 total attempts. On a comparable compute budget, an RL soft actor-critic (SAC) [37] baseline never completes a flip.

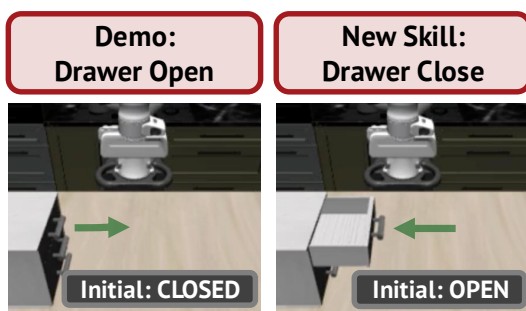

Fig. 4: **Drawer closing.** A VLA is trained only on *open-drawer* demos (left). Closing the drawer (right) requires a new *push drawer closed* primitive executed from an open drawer, which is an OOD initial state for the base policy. INSIGHT can use a VLM completion check to terminate the known approach primitive and trigger the new push drawer primitive.

### B. Simulation: Drawer Closing from Only Drawer Opening Human Demos

The base policy is trained only on *drawer-opening* demonstrations, and INSIGHT must acquire the missing *push drawer closed* primitive starting from an open drawer (Figure 4). We use 50 human teleoperated drawer-opening demonstrations, segmented into three primitives: (1) *move gripper to the drawer handle*, (2) *close gripper*, and (3) *pull the drawer open*. This setting introduces an out-of-distribution (OOD) initial state: the first primitive (the approach) was trained only with the drawer initially closed, but the drawer closing begins with the drawer already open. When executing from this OOD state, the approach primitive may partially or fully close the drawer, since the policy generalizes enough to approach the handle but the learned progress degrades outside the training distribution. To handle this distribution shift, we use a VLM completion check that periodically queries whether the current primitive is complete.

**Even from an OOD initial state, INSIGHT still acquires the missing primitive, using a VLM completion check to bridge imperfect primitive termination.** The VLM completion check terminates the approach primitive and triggers the push-drawer-closed primitive. Across 82 episodes, IN-SIGHT produces 70 successful close-drawer primitives, where incorrect axis selection is the dominant primitive acquisition failure mode. We then retrain a unified VLA using the original drawer-opening demonstrations with the new 70 close-drawer primitives. The final VLA closes the drawer with 100% success over 25 evaluation trials, while retaining its ability to open it. The VLM completion check does not always align exactly with the semantic boundary between moving to the handle and pushing the drawer closed, but it provides a sufficient transition signal for reliable closed-loop drawer closing.

### C. Real-World: Skill Acquisition and Compositional Reuse

We train a VLA on primitives automatically segmented from 50 human pick-and-place demonstrations (grasping the bottle from the top and side), and evaluate INSIGHT on three tasks

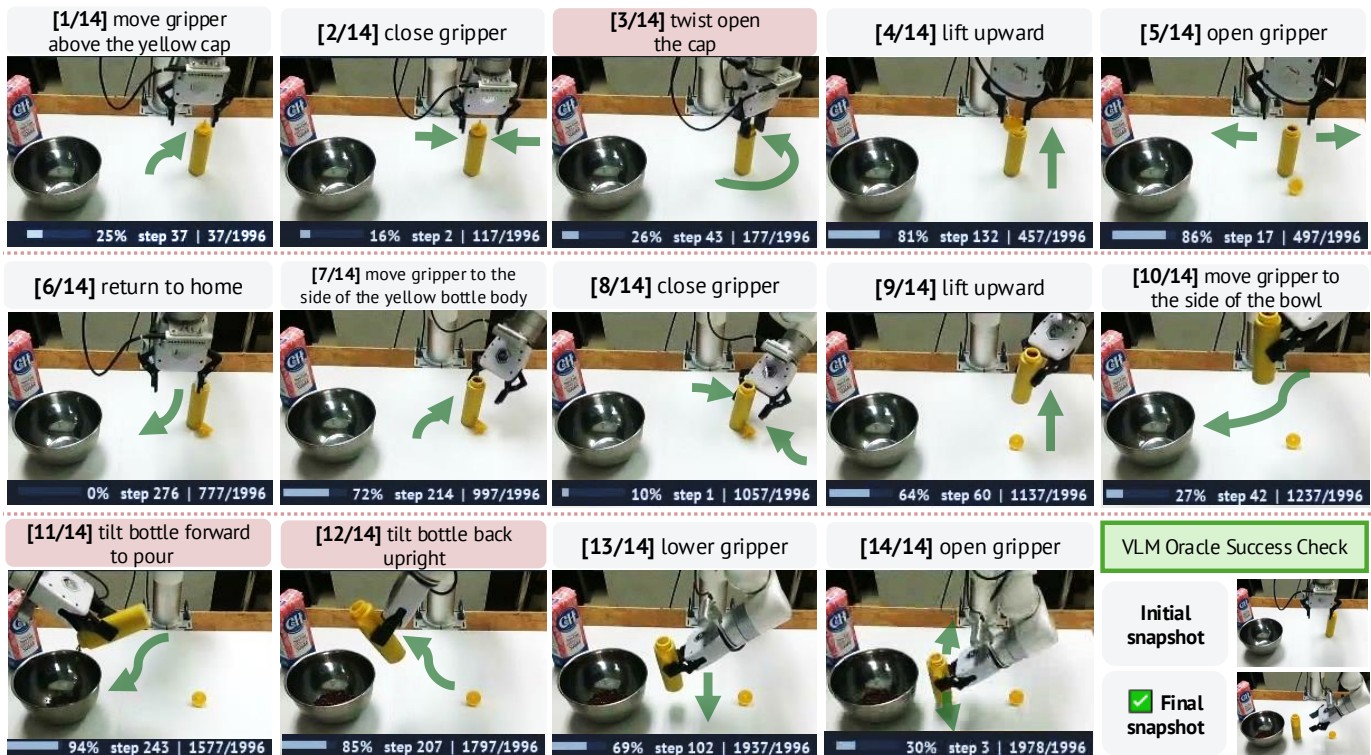

Fig. 5: **Compositional twist-then-pour evaluation rollout.** INSIGHT chains 14 primitives from the separately acquired twist and pour skills, with no end-to-end demonstrations of the combined task. Shaded headers mark primitives acquired autonomously by INSIGHT and added back into the VLA's vocabulary; unshaded primitives are already known from the pick-and-place base demonstrations. The step/progress value shown in each panel is the learned *per-primitive* progress channel, which resets at the start of each primitive rather than increasing monotonically across the full 14-primitive sequence. A VLM oracle verifies task success.

that require primitives absent from this data: twisting open a bottle cap, pouring its contents into a bowl, and a long-horizon task that composes the twisting and pouring skills. We compare against CaP-X [14], which represents the class of methods in which a VLM composes motions to perform new tasks zero-shot at test time without expanding the learned policy, as well as $\pi_{0.5}$, a fine-tuned policy with only human demonstrations and no new primitives from INSIGHT.

**Per-primitive reliability leads to high end-to-end success.** Figure 6 reports per-primitive success rates for the real-world twist, pour, and composition tasks. INSIGHT maintains high success at every primitive, so per-primitive reliability compounds into 92% and 96% end-to-end success on twist and pour, against 32% and 16% for CaP-X and 0% for the $\pi_{0.5}$ baseline, which fails entirely on the twist and pour primitives because of a lack of demonstration data for twisting and pouring. The gap widens on the longer-horizon composition, where INSIGHT reaches 80% end-to-end success while CaP-X drops to 4%.

**Primitives can be composed into new long-horizon skills for continual learning**. As shown in Figure 5, the robot must (1) grasp the bottle from the top, (2) twist to open the cap, (3) re-grasp the bottle from the side, and (4) pour beans into a bowl. This task shows compositional generalization: INSIGHT

TABLE I: **Primitive-acquisition statistics on xArm.** Cost of autonomous data acquisition to reach 20 successful acquired primitives; *Trials* is the total number of episodes. *Robot*, *VLM*, and *Wall* are cumulative robot-motion, VLM-call, and wall-clock time in minutes, with per-trial means (in seconds) in parentheses.

| Skill | Trials | Robot | VLM | Wall |
|-------|--------|-------|-----|------|
| Twist | 23 | 23.8 (62 s) | 8.4 (21.9 s) | 39.7 (104 s) |
| Pour | 31 | 49.6 (96 s) | 26.9 (52 s) | 85.3 (165 s) |

must chain primitives from two different acquired skills, without having end-to-end demonstrations of the combined task. INSIGHT sequences 14 primitives into a successful rollout, reaching 80% end-to-end success compared to 4% for CaP-X.

**INSIGHT improves on execution efficiency compared to Code-as-Policies frameworks that perform each skill zero-shot**. Table I reports the autonomous data acquisition cost per new real-world skill, while Figure 7 breaks down per-trial timing into VLM-call latency and robot execution for INSIGHT and CaP-X. **Each new skill is also efficient to acquire**: Table I shows 20 successful primitives acquired in 23 trials for twisting and 31 for pouring.

**INSIGHT preserves existing skills when new primitives**

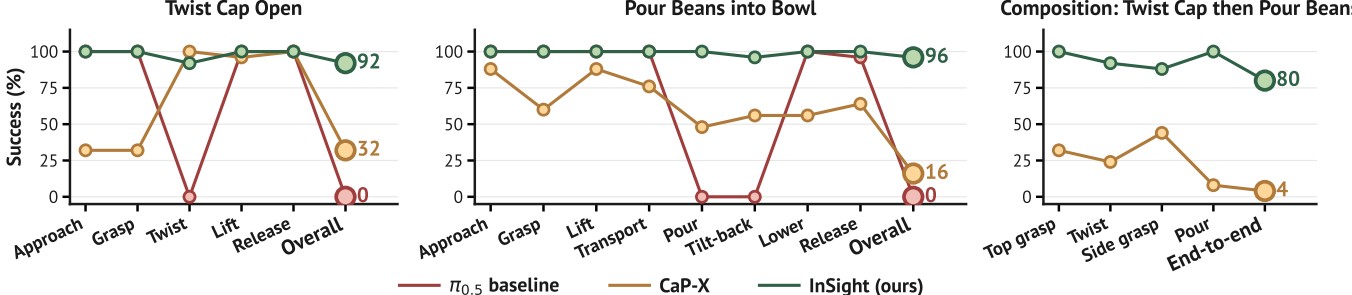

Fig. 6: **Real-world per-primitive success rates**, 25 trials per method. Each marker is the success rate of the labeled primitive across rollouts; *Overall / End-to-end* is full-task success. The $\pi_{0.5}$ baseline is fine-tuned on 50 human pick-and-place demos; INSIGHT additionally uses 20 successful acquired primitive episodes. In the cap twisting (**left**), bottle pouring (**center**), and twist-then-pour (**right**) tasks, INSIGHT consistently outperforms CaP-X [14].

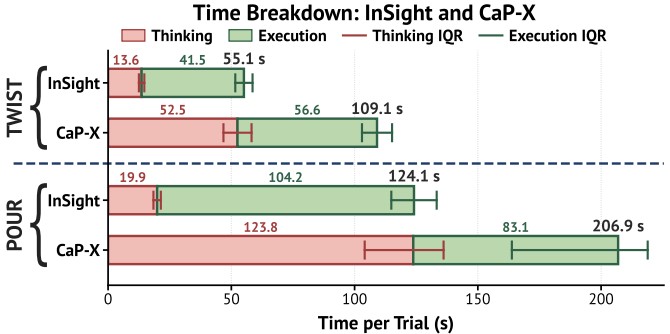

Fig. 7: **Per-trial time decomposition.** INSIGHT spends substantially less wall-clock time per trial than CaP-X on both skills. *Thinking* reports VLM-call latency; *Execution* reports robot motion. $N=25$ evaluation trials per method.

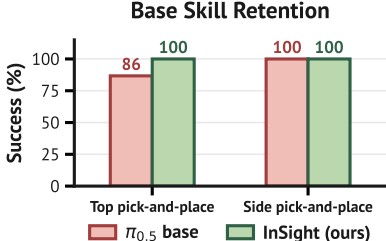

Fig. 8: **Base skill retention.** The unified VLA retains the original pick-and-place skills after adding twist and pour primitives ($N=15$).

**are added.** As shown in Figure 8, even after adding the newly acquired twist and pour primitives, the unified VLA retains 100% success on the original top- and side-pick-and-place skills.

### D. Real-World: Sweeping from Only Scooping Human Demos

**INSIGHT extends beyond grasping to contact-rich, non-prehensile motions.** This final task returns to our motivating scenario: a robot on Mars trained only to scoop rocks must sweep debris off its panels, with no sweeping demonstrations available. We evaluate whether INSIGHT can acquire a *sweep-*

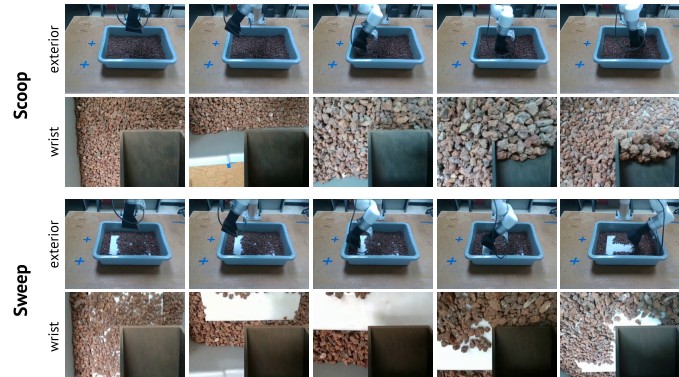

Fig. 9: **Sweeping from only scooping human demonstrations.** Exterior and wrist views of the demonstrated scooping skill (top) and the sweeping skill acquired through INSIGHT (bottom). Since both scooping and sweeping require the gripper to be lowered to the rocks, INSIGHT acquires sweeping by adding a lateral-push primitive to the scooping primitives.

*ing* primitive from demonstrations of scooping. Scooping and sweeping share the same approach and lowering primitives and differ only in the final contact motion. While scooping rocks out of the bin requires the gripper to descend below the rocks, sweeping requires a lateral push across the surface, as shown in Figure 9. The VLM flags this lateral push as the missing primitive. INSIGHT acquires the sweeping primitive autonomously and succeeds in all $5/5$ evaluation trials.

### V. CONCLUSION, LIMITATIONS, AND FUTURE WORK

We present INSIGHT, a method for autonomous skill acquisition in VLAs through VLM-guided primitive gap discovery and execution. By training on autonomously segmented primitives, identifying primitive gaps via VLM reasoning, and generating training data through VLM-guided low-level control, INSIGHT enables robots to acquire new skills without additional human demonstrations.

Our work suggests several directions for future work. First, skill gap execution is restricted to single-axis motions, limiting the complexity of acquirable primitives. Future work would acquire richer primitives using VLM-generated waypoints,

trajectory optimization, or online RL. Second, INSIGHT learns only from successful rollouts and discards failed attempts; incorporating failure analysis and VLM feedback [38] to learn from these failures would improve sample efficiency. INSIGHT also depends on the VLM for segmentation, planning, primitive-gap proposal, and success verification, so VLM errors can propagate into the acquired primitives; validating accepted rollouts with independent or redundant checks is a promising direction. In addition, human environment resets are still necessary in this work, as each rollout requires manual resets. One way to mitigate this cost is by exploring rollouts with real-to-sim-to-real pipelines or a learned world model [39, 40] that can test candidate rollouts before real-world execution, which can be especially crucial in safety-critical settings. Lastly, INSIGHT can be extended to embodiments with higher degrees of freedom, such as mobile manipulators or humanoids.

### ACKNOWLEDGMENTS

The authors thank Joseph Bowkett and Daniel Pastor for valuable discussions and for providing the 3D printed scooper for the xArm. M. Wang is supported by the NASA NSTGRO Fellowship. S. Tian was supported by NSF GRFP Grant No. DGE-1656518.

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

# APPENDIX A
## IMPLEMENTATION DETAILS

We use the $\pi_{0.5}$ VLA [2] in our experiments, although INSIGHT is agnostic to the underlying VLA. We fine-tune with LoRA [35] (Gemma-2B backbone and Gemma-300M action expert with other weights frozen), with each segmented primitive as a separate training episode conditioned on its primitive label as the language prompt. The policy takes two $224 \times 224$ RGB views (scene and wrist) and the end-effector pose and gripper state, and outputs end-effector deltas, an absolute gripper command, and a learned progress channel $\in [0, 1)$ supervised with the normalized timestep within each primitive segment. A primitive terminates when the progress channel exceeds a threshold (typically 0.95), when end-effector motion falls below an auto-advance threshold, or (for out-of-distribution "move to" primitives) when a VLM completion check fires.

# APPENDIX B
## VLM DEMONSTRATION SEGMENTATION, PLAN, PRIMITIVE GAP PROPOSAL, AND ORACLE CHECK

INSIGHT queries a vision-language model (Gemini 3 Flash [41]) in four roles, each constrained to return strict JSON: (i) offline **segmentation** of human demonstrations into primitive-labeled trajectories (B-A); (ii) **planning**, which decomposes a novel task into a primitive sequence and flags primitive gaps (B-B); (iii) **primitive-gap proposal**, which maps each gap to a single-axis motion the low-level controller executes (B-C); and (iv) **oracle** checks that verify primitive- and task-completion from images (B-D). The system prompts are reproduced verbatim below ({...} are runtime-filled fields).

### A. Demonstration Segmentation

Demonstrations are segmented offline in three stages. First, the VLM decomposes the task instruction into an ordered primitive sequence. Second, the subsampled video is passed frame-by-frame and each frame is assigned to a plan primitive, cross-checking the image against a per-frame end-effector motion caption that reports the dominant translation/rotation axis, then returns the boundary frames between consecutive primitives. Each frame caption has the form:

---

**Per-frame motion caption (segmentation input)**

```
Frame 120 | EE x=+0.213 y=-0.041 z=+0.118 rx=+3.04 ry=-0.02 rz=+1.57
| d dx=+0.004 dy=-0.001 dz=-0.012 |dxy|=0.004 |dz|=0.012
|drxy|=0.01 |drz|=0.00 dom=-z
```

---

Third, each boundary is refined by a localized pass that reconciles the end-effector delta change-point with the earliest visually unambiguous frame. The result is a set of contiguous, primitive-labeled segments, each of which becomes one training episode (Appendix A).

### B. Task Planning and Primitive-Gap Flagging

Given a goal and the current primitive vocabulary, the planner returns the full primitive sequence, an execution note per step, and the subset of steps that are novel (primitive gaps; the `skill_gaps` field below). Each primitive gap is constrained to a single-axis motion, so a multi-motion task yields multiple gaps.

---

**System prompt: PLAN_TASK**

You are a robot task planner. Scene: {scene_context}

AVAILABLE PRIMITIVES (each is general-purpose and adapts to context): {primitives}

RULES:
1. Break the goal into fine-grained steps. Use existing primitives for every sub-step they cover -- a skill gap should only be the novel part, not a bundle of existing + novel actions.
2. Only create a skill gap when the desired outcome is fundamentally different from what any existing primitive produces. If an existing primitive could achieve the same result (even if executed differently), use it and put execution details in step_notes instead.
3. Every step goes in primitive_sequence -- including new ones.
4. New primitives also go in skill_gaps (must appear in BOTH lists).
5. Name new primitives by their desired EFFECT, not the robot motion.

---

6. For each step, add a note on execution (approach, grasp, how it enables the next step).
7. After the final step, the runtime returns the gripper to a safe home pose, so the gripper does not need to be cleared from the workspace by a final step in the plan. Each step should make a distinguishable contribution to the goal -- avoid adding a final step whose only effect is repositioning the gripper.
8. Each skill gap is one single-axis motion (one translation OR one rotation along one axis, in one direction). If the goal involves multiple distinct motions, create a separate skill gap for each.

Example 1 -- pick and place (all existing, no skill gaps):
primitive_sequence: ["move gripper to the red lego block", "close gripper", "lift upward", "move gripper to target", "lower gripper", "open gripper"]
skill_gaps: []

Example 2 -- inserting an object (one new skill gap):
primitive_sequence: ["move gripper to object", "close gripper", "lift upward", "move gripper to target", "insert object into slot", "open gripper"]
skill_gaps: ["insert object into slot"]

Respond with ONLY valid JSON:
{"primitive_sequence": ["step1", "step2", ...],
  "step_notes": ["execution note for each step"],
  "skill_gaps": ["new primitives not in available list"],
  "reasoning": "brief explanation",
  "confidence": 0.0-1.0,
  "requires_new_primitive": true or false}

**Example output** (pour data collection; the planner detects two rotational gaps):

---

**Example output: PLAN_TASK**

{"primitive_sequence": ["move gripper to the side of the bottle",
  "close gripper", "lift upward", "move gripper to the side of the bowl",
  "pitch bottle forward to pour", "tilt bottle back upright",
  "lower gripper", "open gripper"],
 "skill_gaps": ["pitch bottle forward to pour", "tilt bottle back upright"],
 "reasoning": "The existing primitives cover linear movement, grasping, and
  lifting. However, the pouring action requires a rotational movement (pitch)
  not in the available list. Per Rule 8, each single-axis motion in one
  direction is its own skill gap, so the forward pitch and the backward tilt
  to upright are two separate gaps.",
 "confidence": 1.0,
 "requires_new_primitive": true}

---

A full executed plan (the twist-then-pour composition) is shown in Appendix C.

### C. Primitive-Gap Proposal

For each flagged gap, the VLM sees the exterior and wrist images and returns a single-axis motion: an axis (dx,dy,dz base-frame translation or drx,dry,drz gripper-local rotation), a signed magnitude in meters or degrees, and an already_complete flag. The controller then drives the arm along that one axis until the completion check (below) fires.

---

**System prompt: PREANALYZE**

Determine the single-axis MOTION (translation OR rotation) needed to achieve the GOAL on a real xArm. The controller drives the arm along whichever single axis you pick.

IMAGES:
- IMAGE 1 (exterior overview): use for TRANSLATION reasoning.
- IMAGE 2 (wrist down-view, moves with the gripper): use for ROTATION
    ↪  axis
  selection.

TRANSLATION (dx, dy, dz) is in the BASE frame:
  +X = forward (into workspace), +Y = left, +Z = up.

---

ROTATION (drx, dry, drz) is in the GRIPPER's LOCAL frame (rotate WITH
    ↪  gripper):
  drz: Axial twist around the gripper's local Z axis (the camera's line of
    sight). Spins the held object in place around its own centerline like a
    screwdriver. It CANNOT pivot, tilt, or invert the angle of an object.
  dry: Pitch rotation around the gripper's local Y axis. Tilts/nods the gripper
    body forward or backward along its opening/closing path.
  drx: Roll rotation around the gripper's local X axis. Tilts the gripper body
    laterally sideways.

To pick the rotation axis: Do NOT pick the axis from the global room frame,
camera frame, or verbs like "sideways/forward" -- look at IMAGE 2, since robot
base frame coordinates differ from gripper coordinates.
- If the object needs to pivot/tilt forward/backward relative to the gripper
  body, pick dry.
- If the object needs to tilt laterally sideways relative to the gripper body,
  pick drx.
- If the object needs to twist/spin in place along its own centerline without
  changing its physical tilt angle, pick drz.

GEOMETRIC CONSTRAINT WARNING:
1. Never select drz for any motion that requires an object to tip over, invert,
   or pivot its top towards a target; drz only spins the object on its own axis.
2. The wrist camera moves with the gripper; its local axes are independent of
   the global room frame. Never select an axis based on where a target appears
   to sit (left, right, up, down) in IMAGE 1. Map the required tilt strictly to
   the local structure of the gripper fingers in IMAGE 2.

BE AWARE: Depth and gripper biases may exist due to the close-up wrist view.

SIGNED MAGNITUDE:
- Translation: meters, typical 0.05-0.20 m.
- Rotation:    degrees, typical 30-180 deg.
- Sign: along the chosen axis.

If already complete, set already_complete=true.

Respond with ONLY valid JSON:
{"current_state": "...",
 "target_state": "...",
 "reasoning": "...",
 "axis": "dx" | "dy" | "dz" | "drx" | "dry" | "drz",
 "signed_magnitude_m": 0.0,
 "signed_magnitude_deg": 0.0,
 "already_complete": false}

Set signed_magnitude_m=0 for rotation; signed_magnitude_deg=0 for
    ↪  translation.

---

**Example outputs** (two resolved gaps from logged runs):

---

**Example output: PREANALYZE — "unscrew bottle cap"**

{"current_state": "The gripper is securely holding the cap of the yellow
  bottle.",
 "target_state": "The cap is rotated 150 degrees counter-clockwise to detach
  it from the bottle threads.",
 "reasoning": "Unscrewing is an axial twist of the held cap about the
  gripper's line of sight (local Z), so drz; ~150 deg detaches it.",
 "axis": "drz",
 "signed_magnitude_m": 0.0,
 "signed_magnitude_deg": 150.0,
 "already_complete": false}

---

**Example output: PREANALYZE — "pitch bottle forward to pour"**

{"current_state": "The gripper is holding the yellow bottle in a side-grasp,
  positioned next to the metal bowl.",
 "target_state": "The yellow bottle is tilted over the bowl so that all its
  contents are poured out.",
 "reasoning": "To pour the contents into the bowl, the robot needs to rotate
  the gripper. Looking at IMAGE 2 (the wrist camera view), the bowl is
  positioned 'below' the bottle. A pitch rotation around the gripper's local
  Y-axis (dry) tilts the gripper body and the held bottle forward/downward
  into the bowl. A magnitude of 110 degrees ensures the bottle is tilted

enough to empty all the beans. drz is excluded by the geometric
constraints because the motion requires tipping the bottle over.",
"axis": "dry",
"signed_magnitude_m": 0.0,
"signed_magnitude_deg": 110.0,
"already_complete": false}

### D. Oracle and Completion Checks

After a plan executes, an *oracle* compares the initial and
final scene images and accepts the trial only if the task is
achieved; accepted trials become training demonstrations and
the rest are discarded.

---

**System prompt: TASK_COMPLETION (oracle)**

You are a vision system inspecting a robot manipulation scene from a fixed
external camera. Compare two images (BEFORE and AFTER a robot action)
↪  and decide
whether the task was completed.

---

paired with the user message:

---

**User message: oracle**

IMAGE 1 = BEFORE the action (gripper at home, scene in initial state).
IMAGE 2 = AFTER the action.

Task: {task}.

Question: Was the task completed?

Respond with valid JSON only:
{"completed": true | false, "reasoning": "<one sentence>"}

---

**Example oracle verdicts** (accepted twist-then-pour trials):

---

**Example output: TASK_COMPLETION**

{"completed": true, "reasoning": "The yellow bottle is uncapped, its
contents have been poured into the bowl, and the bottle has been
returned to the table with the gripper now open."}

{"completed": true, "reasoning": "The bottle has been successfully uncapped,
moved from its initial position to the bowl, and released, and the gripper
has returned to an open state at the home position."}

---

**System prompt: PRIMITIVE_DONE (completion
check)**

Determine if the robot primitive has been completed.

You receive two images:
- IMAGE 1 (exterior camera): side view across the table. PRIMARY signal for
  depth and height.
- IMAGE 2 (wrist camera): top-down from the gripper. Use ONLY for
  centering/identification.

CRITICAL: Top-down views (IMAGE 2) make objects appear close even when
  ↪  there is
a large vertical gap. ALWAYS judge vertical proximity from IMAGE 1 (side
  ↪  view).
If you can see ANY visible vertical gap between the gripper bottom and the
target in IMAGE 1, the primitive is NOT done -- even if IMAGE 2 shows them
overlapping.

For "move gripper to X" or "touch X":
- Done = gripper bottom is contacting X or within ~5mm of X in IMAGE 1.
- NOT done = visible vertical gap between gripper bottom and target in IMAGE
  ↪  1.

For state-change primitives (open/close/push/pull/rotate/scoop/sweep/lift):

---

- Done = the target state is visibly achieved (drawer closed, object pushed
  away, rocks displaced, etc.) in IMAGE 1.

Reasoning must explicitly describe what you see in IMAGE 1, not just IMAGE
  ↪  2.

Respond with ONLY JSON: {"done": true or false,
  "reasoning": "brief, references IMAGE 1"}

---

APPENDIX C
VLM-GENERATED PLANS

The following is an example VLM-generated plan for
the twist-then-pour compositional task. Note that all of the
primitives are marked as *known* because they have all been
added to the VLA, including the primitives generated through
INSIGHT (*twist open the cap*, *tilt bottle forward to pour*, *tilt
bottle back upright*).

---

**Primitive Plan for Twist-then-Pour**

**Task:** Open the yellow bottle cap, regrasp the bottle, and pour its contents into
the bowl.

1) `KNOWN` **move gripper above the yellow bottle cap** — Move the gripper
   into a top-down approach position centered over the yellow cap.
2) `KNOWN` **close gripper** — Close the gripper to secure a firm grasp on
   the cap.
3) `KNOWN` **twist open the cap** — Perform a 180-degree counterclockwise
   rotation to unscrew the cap from the bottle.
4) `KNOWN` **lift upward** — Lift the cap vertically to ensure it is completely
   detached from the bottle threads.
5) `KNOWN` **open gripper** — Open the gripper to drop the detached cap
   onto the workspace.
6) `KNOWN` **return to home** — Execute the mandatory hardware reset to
   return the robot to its canonical home pose.
7) `KNOWN` **move gripper to the side of the yellow bottle body** — Move
   the gripper to a side-approach position relative to the bottle body.
8) `KNOWN` **close gripper** — Close the gripper to perform a side grasp on
   the now-uncapped bottle.
9) `KNOWN` **lift upward** — Lift the bottle upward to clear the table for
   movement.
10) `KNOWN` **move gripper to the side of the bowl** — Transport the bottle
    to the side of the bowl in preparation for pouring.
11) `KNOWN` **tilt bottle forward to pour** — Tilt the bottle forward over the
    bowl to empty its contents.
12) `KNOWN` **tilt bottle back upright** — Rotate the bottle back to a vertical,
    upright orientation.
13) `KNOWN` **lower gripper** — Lower the bottle back down to the table
    surface.
14) `KNOWN` **open gripper** — Open the gripper to release the bottle.