# OpenReview forum: "InSight: Self-Guided Skill Acquisition via Steerable VLAs"
_roboticsfoundation.org/RSS/2026/Workshop/RL4VLA — RL4VLA_

### Official Review · Reviewer_DDz7 · 2026-06-27

**Rating:** 7
**Confidence:** 4

**Review:**

## Summary

This paper presents InSight, a framework for self-guided skill acquisition in VLA policies. The system first converts human demonstrations into primitive-labeled segments and fine-tunes a VLA to be steerable by primitive labels. Given a new task, a VLM identifies missing primitives, proposes single-axis low-level motions for those gaps, filters successful rollouts with a VLM oracle, and retrains the VLA so that the newly acquired primitives become reusable. The paper demonstrates the method on simulation tasks and real xArm manipulation tasks, including twisting, pouring, sweeping, and a long-horizon twist-then-pour composition.

## Strengths

- The motivation is practical, and the proposed skill acquisition method is intuitive.
- The real-robot results are promising. The twist, pour, and 14-primitive composition experiments show that the learned primitives can be reused compositionally.

## Weaknesses

- The system is not fully autonomous, since human environment resets are still required after rollouts. It would be useful to discuss whether this limitation can be improved in future work.
- The primitive definition keeps each skill simple enough for the VLM to produce reasonable action parameters. However, for long-horizon, complex, and fine-grained manipulation tasks, this definition may limit the robot’s ability to learn and execute more complex skills.

---

### Official Review · Reviewer_NhmW · 2026-06-28
**Review of paper "InSight: Self-Guided Skill Acquisition via Steerable VLAs"**

**Rating:** 5
**Confidence:** 4

**Review:**

Summary

The paper proposes INSIGHT, an autonomous skill‑acquisition pipeline for Vision‑Language‑Action systems that combines an automated demonstration segmentation stage with a VLM‑guided data flywheel. The segmentation stage decomposes demonstrations into labeled primitives using VLM plan decomposition and end‑effector pose cues to enable primitive steerability. The flywheel identifies missing primitives for a novel task, attempts autonomous demonstrations using VLM‑proposed low‑level control, and automatically labels and integrates successful trials into the VLA training set. Experiments in simulation (7DoF Franka Panda in LIBERO) and on hardware (6DoF UFactory xArm) evaluate block flipping, drawer closing, bottle twisting and pouring, long‑horizon composition, skill retention, and contact‑rich sweeping. Results show strong gains from autonomously acquired primitives (e.g., up to 96% pouring success, 80% on a 14‑primitive long task) while retaining base skill performance.

Strengths
1. This work is strong alignment with workshop themes such as VLM fine‑tuning for robotics and hierarchical multimodal planning.
2. The two‑stage design (segmentation then flywheel) is conceptually clean and directly addresses scalable, autonomous primitive acquisition without target‑skill human demos.
3. Validation across simulation and real hardware, including OOD initial states, compositional reuse, and contact‑rich motions, increases ecological validity.

Weaknesses
1. The paper reports aggregate success rates but lacks systematic ablation studies isolating the contributions of VLM plan decomposition, end‑effector pose heuristics, and success‑filter thresholds, weakening causal claims.
2. Random seeds, dataset splits, success criteria, prompt templates, and hyperparameters are not fully documented, which hinders reproducibility and fair comparison.
3. It is not shown how performance scales as the primitive library grows, how often the flywheel proposes redundant or low‑value primitives, or how well the approach generalizes to substantially different object geometries or dynamics.

Questions for the Authors
1. How often does the flywheel propose infeasible or unsafe primitives, and what safeguards (sim‑to‑real checks, conservative controllers) are in place?
2. How sensitive are results to the VLM prompt design, success‑thresholds for automatic labeling, and number of autonomous attempts per missing primitive?

---

### Decision · Program_Chairs · 2026-07-03

**Decision:**

Accept

**Comment:**

The paper presents a method for autonomous skill acquisition for vision-language-action models, validated in both simulation and real robot experiments. The reviewers found the approach novel and the results promising. The main concerns are limited reproducibility details and practical limitations, such as manual resets and the simplified primitive design. The paper also presents a comparison with RL finetuning of VLAs, therefore, we believe the paper is a valuable contribution to the workshop. For the camera-ready version, the authors should add key implementation details to improve reproducibility and briefly discuss the current limitations, while leaving more extensive analysis and improvements as future work.